# Adjunctive Bright Light Therapy for Non-Seasonal Major Depressive Disorder: A Randomized Controlled Trial

**DOI:** 10.3390/ijerph191912430

**Published:** 2022-09-29

**Authors:** I-Peng Chen, Chun-Chao Huang, Hui-Chun Huang, Fan-Pei Gloria Yang, Kai-Ting Ko, Yun-Tse Lee, Fang-Ju Sun, Shen-Ing Liu

**Affiliations:** 1Department of Psychiatry, MacKay Memorial Hospital, Taipei 104217, Taiwan; 2Department of Medicine, MacKay Medical College, New Taipei City 252005, Taiwan; 3Department of Radiology, MacKay Memorial Hospital, Taipei 104217, Taiwan; 4MacKay Junior College of Medicine, Nursing and Management, Taipei 112021, Taiwan; 5Department of Medical Research, MacKay Memorial Hospital, Taipei 104217, Taiwan; 6Department of Foreign Languages and Literature, National Tsing Hua University, Hsinchu 300044, Taiwan; 7Center for Cognition and Mind Sciences, National Tsing Hua University, Hsinchu 300044, Taiwan; 8Department of Radiology, Graduate School of Dentistry, Osaka University, Osaka 565-0871, Japan; 9Institute of Biomedical Informatics, National Yang Ming Chiao Tung University, Hsinchu 300093, Taiwan

**Keywords:** light therapy, light treatment, depression, major depressive disorder

## Abstract

This double-blind, randomized controlled trial assessed bright light therapy (BLT) augmentation efficacy compared with placebo light in treating non-seasonal major depressive disorder. The study participants belonged to a subtropical area (24.5°–25.5°N) with extensive daylight and included outpatients who had received stable dosages and various regimens of antidepressive agents for 4 weeks before enrollment. The outcomes were the 17-item Hamilton Depression Rating Scale, Montgomery–Asberg Depression Rating Scale, and Patient Health Questionnaire-9, which were assessed at weeks 1, 2, and 4. A total of 43 participants (mean age 45 years, ranging from 22–81) were randomized into the BLT [*n* = 22] and placebo light groups [*n* = 21]. After a 4-week administration of morning light therapy (30 min/day), depressive symptoms did not reduce significantly, which might be due to the small sample size. Nonetheless, this study had some strengths because it was conducted in warmer climates, unlike other studies, and examined diverse Asians with depression. Our findings suggest that several factors, such as poor drug response, different antidepressive regimens, duration of BLT, and daylength variability (i.e., natural daylight in the environment) may influence the utility of add-on BLT. Researchers may consider these important factors for future non-seasonal depression studies in subtropical environments.

## 1. Introduction

According to the World Health Organization, major depressive disorder (MDD) is one of the leading causes of disability worldwide. Many patients with MDD respond poorly to antidepressants. Specifically, only one-third of patients achieve remission from depression after first-line antidepressant therapy, one-half achieve remission after two lines of antidepressant treatments, and approximately two-thirds of all patients achieve remission after four lines of antidepressant therapy [1]. Following an inadequate response to an antidepressant for MDD, clinicians will next consider several other strategies, including the use of a different antidepressant, combined or augmented treatment with other antidepressants, or the use of non-antidepressant agents or non-pharmacological treatment [2,3]. Bright light therapy (BLT) is one option for augmentation.

BLT has been recommended by the American Psychiatric Association for the treatment of seasonal affective disorder (SAD) because of its established efficacy [4]. BLT has the potential advantage of being effective in modifying circadian phase-shift, sleep, sympathovagal balance, and alertness [5]. In addition, increasing studies have investigated BLT in the treatment of non-seasonal MDD. In a recent meta-analysis [6], light therapy had a mild-to-moderate statistical effect on depressive symptoms compared to those in a control group of patients with non-seasonal depression. The tendency toward therapeutic efficacy revealed that a single treatment of light therapy was better than adjunctive light therapy. Efficacy was greater in outpatients than in inpatients. However, the quality of overall evidence is low, and the efficacy of light therapy for non-seasonal MDD has varied internationally. In addition, few studies were conducted in Asia [6,7]. Additional well-designed studies are needed to confirm the efficacy of BLT for treating non-seasonal depression.

Regarding the augmentation effect of BLT, few studies used a randomized controlled trial (RCT) design to evaluate the efficacy of BLT combined with different antidepressive regimens in patients with MDD. In a meta-analysis examining light therapy and antidepressant drugs alone and in combination, most studies used selective serotonin reuptake inhibitors (SSRIs, 6 over 7), with only one trial using tricyclics [8]. The applicability of BLT augmentation to various antidepressive agents in actual clinical practice has not been clarified. Meanwhile, the reproducibility of adjunctive BLT in subtropical climates with more daylight is uncertain.

Hence, this study primarily aimed to examine the efficacy of 4-week BLT augmentation in the treatment of patients with non-seasonal unipolar MDD in Taipei, Taiwan, which is situated in a subtropical area (24.5°–25.5° N). Antidepressive therapy was maintained for 4 weeks prior to this trial to generate a homogeneous sample at baseline. We used a randomized, controlled design to compare the antidepressive effects of BLT and dim red light (DRL) as a placebo. It was hypothesized that BLT combined with antidepressive agents would more effectively reduce depressive symptoms than placebo light and antidepressive agents.

## 2. Materials and Methods

### 2.1. Study Protocol

The study design was a double-blind RCT for 4 weeks. We conducted this trial at MacKay Memorial Hospitals, Taipei, Taiwan from June 2019 to June 2020. All participants provided written informed consent after the study procedures had been explained. The Institutional Review Board (IRB) at MacKay Memorial Hospital approved the protocol (IRB number: 18MMHIS114e). It was registered at ClinicalTrials.gov in May 2019 (identifier: NCT 03941301).

### 2.2. Eligibility Criteria

Participants were recruited through referrals from psychiatrists’ outpatient clinics. All diagnoses were confirmed by board-certified psychiatrists in accordance with Diagnostic and Statistical Manual of Mental Disorders, fifth edition (DSM-5). The inclusion criteria were as follows: age ≥20 years, current DSM-5 diagnosis of MDD, current episode lasting for 6 weeks or greater, 17-item Hamilton Depression Rating Scale (HAMD-17) score ≥13, and use of antidepressants at stable dosages for at least 4 weeks prior to enrollment. The dosages of antidepressants should adhere to the recommendations of the relevant guidelines [9,10,11,12]. Furthermore, participants and the treating psychiatrists were asked to not change the medications and dosages during the treatment phase of the study. No types of antidepressants were restricted. Comorbid anxiety disorders were allowed. The exclusion criteria were as follows: SAD; any manic, hypomanic, or mixed episodes; psychotic disorder; any alcohol or substance use disorder in the past 30 days; intellectual disability; dementia; cognitive impairment; organic brain syndrome; chronic eye diseases; severe illness that could require hospitalization in the near future; treatment with photosensitizing drugs, and; photosensitive epilepsy or migraine.

### 2.3. Procedures

For participants meeting the eligibility criteria, the research assistants recorded their age, gender, marital status, highest level of education, employment status, age at onset of MDD, number of depressive episodes, pharmacologic treatment and dosages, and time of month of enrollment. Research assistants assessed clinical symptoms using HAMD-17 and the Montgomery–Asberg Depression Rating Scale (MADRS). The participants were asked to complete Patient Health Questionnaire-9 (PHQ-9), Morningness–Eveningness Questionnaire: Self-Assessment Version (MEQ-SA) [13], an adverse events scale, and an expectation scale. MEQ-SA was used to obtain the chronotype of the circadian rhythm, which is associated with the response to light therapy. The chronotype was classified as morning/intermediate or evening type based on the MEQ-SA score. Specifically, scores ≤ 41 indicated evening type, which reflected the delayed sleep-wake phase [14]. The adverse events scale was used to monitor the side effects of light therapy. The expectations of participants can affect outcomes on BLT; thus, they were rated using a brief multipoint rating scale (1–10) at baseline. A score of 10 indicated full expectancy of treating depression with light therapy, whereas a score of 1 indicated no expectancy.

### 2.4. Randomization and Blinding

The randomization scheme was generated using a computer. An independent researcher concealed the sequence and assigned the interventions. The participants were randomly assigned in a 1:1 ratio to BLT or DRL with a block size of six. The non-blinded personnel dispensed the study light boxes and trained the participants on their use. These personnel worked separately from the research assistants who conducted visits and performed the clinical ratings. The raters and medication-prescribing psychiatrists were all blinded to the allocated treatment.

### 2.5. Intervention and Outcome Assessments at Weeks 1, 2 and 4

We used HAPPYLIGHT LUCENT VT22 (Verilux Company, Waitsfield, Vermont, US) with the dimensions of 6.5 inches (W) × 8.5 inches (H) × 4.5 inches (D) as the bright light unit, which could emit 10,000 lux of white and ultraviolet-free full-spectrum light. The white light-emitting diode light had a color temperature of 5000 Kelvin. The red-light unit was used to deliver 70 lux of DRL, which has been frequently used as placebo light in trials [6]. The appearance of the placebo light unit was similar to that of the bright light unit. All the light boxes were packed in the same cardboard boxes. Participants were provided standardized verbal and written instructions on the use of the light box. They were instructed to sit in front of the light box and place it on a desk at a distance of approximately 40 cm from the eyes and to avoid looking at it directly. Participants were asked to begin light therapy with 30-min sessions after being awake before 09:00 a.m. at home for 4 weeks. They were assessed for outcome measures and adverse events at weeks 1, 2, and 4. To monitor adherence to light therapy, participants were required to record their sessions using a light box on the self-report adherence form and provide the start and finishing times for each of the study days. The adherence form was returned to the research assistants at each visit.

### 2.6. Outcome Measures

#### 2.6.1. HAMD-17 and Its Subscales

HAMD-17 is a 17-item scale that assesses the overall levels of severity of depressive symptoms and responses to treatment [15]. It also emphasizes psychic anxiety and somatic anxiety. Each item is scored from 0 to 4 or 0 to 2. Higher scores indicate greater severity, and the total score ranges 0–52. The severity ranges for MDD on HAMD-17 are mild (8–16), moderate (17–23), and severe (≥24) [16]. Good inter-rater reliability and internal reliability have been found in the Chinese version of HAMD-17 [17]. The scores of the full HAMD scale were divided into the following six subscales: Bech melancholia scale [18], Maier–Philipp severity subscale [19], Gibbons global depression severity scale [20], Santen scale [21], anxiety subscale [22], and retardation subscale [23] (Appendix A).

#### 2.6.2. MADRS

MADRS was developed to assess changes in the severity of depression [24]. Higher scores suggest increasing severity of depression, and the total score ranges from 0–60. MADRS consists of ten items, and each item is scored from 0 to 6. It does not address somatic anxiety. The Chinese version has been broadly used in studies.

#### 2.6.3. PHQ-9

PHQ-9 consists of nine items evaluating the nine DSM criteria of MDD over the past 2 weeks [25]. This self-rating scale does not focus on symptoms of anxiety. Each item is rated from 0 to 3, giving a total score of 0–27. Higher scores reflect a higher likelihood of MDD. The Chinese version of PHQ-9 has been validated for primary care adults in Taiwan [26]. PHQ-9 scores of 10 or higher displayed a sensitivity of 0.86 and specificity of 0.94 for detecting MDD.

### 2.7. Primary and Secondary Outcomes

The primary outcomes were total scores of HAMD-17 at weeks 1, 2, and 4. The secondary outcomes included the change of the HAMD-17 total score from baseline to different visits, the rate of response (proportion of patients with ≥50% reductions of the HAMD-17 score) [27], and the rate of remission (defined as HAMD-17 score ≤ 7) [28].

Moreover, the total scores and changes of MADRS and PHQ-9 scores, the rate of response (proportion of patients with ≥50% reductions of the MADRS score and PHQ-9 score), and the rate of remission (defined as MADRS score≤ 10) were recorded [29]. The six subscales of HAMD-17 were analyzed. The consecutive ratings of the adverse events scale were evaluated at each visit.

### 2.8. Definition of the Subgroup

Our sample was situated in a subtropical area (24.5°–25.5° N). Daylength (duration from sunrise to sunset) was used as the parameter of natural daylight [30,31], and the mean daylength was 12h: 9m (SD 1:04, range 10:35–13:42) from June 2019 to May 2020, and there were less seasonal variations than in areas of higher latitudes. The climate data were obtained from the Central Weather Bureau official website (https://www.cwb.gov.tw/). The participants were divided into subgroups using a cutoff of 12 h/day. A natural daylight duration of <12 h/day indicated when individual participants started to receive light therapy in the months of October to March.

### 2.9. Measures of Side Effects

The side effects of light therapy were measured using the adverse events scale [32,33] at each visit. The scale consists of potential adverse effects that subjects might experience during light therapy, including headache, dizziness, blurred vision, dry mouth, nasal congestion, nausea, vomiting, palpitation, chest tightness, dysuria, constipation, diarrhea, sleep problem, fatigue, drowsiness, agitation, irritability, nervousness, anxiety, excitement, skin discomfort, poor appetite, increase appetite, body weight gain, body weight loss, and sexual desire changes. Each item is rated according to severity (absent, mild, moderate, or severe) in relation to the light therapy.

### 2.10. Statistical Analysis

Power analysis: The original sample size was calculated using the software (G*Power 3.1). Medium effect size estimates were based on a meta-analysis that demonstrated overall medium effects of BLT on non-seasonal depression [34]. To achieve a 95% two-sided confidence level, 80% power, and a medium effect size (Cohen’s *f*) of 0.25 [35] between the treatment and control groups, the target sample size was determined to be 41 participants in each arm. However, before the target sample size was achieved, the recruitment had to be discontinued owing to a COVID-19 outbreak, slow enrollment process, and funding expiration.

The analysis was conducted on an intent-to-treat basis, including all participants who were allocated to treatment. An independent-samples t-test (or the Mann–Whitney U test for skewed data) for continuous variables and the chi-squared test (or Fisher’s exact test for small proportions) for categorical variables were applied to assess differences in the baseline characteristics and outcome measures between the BLT and DRL groups. Non-parametric correlations were calculated using the Spearman rank correlation coefficient. We assessed the normality of distribution using the Kolmogorov–Smirnov or Shapiro–Wilk tests or graphical methods.

Hierarchical linear mixed-modeling (HLM) can detect differences between treatment groups at baseline and over time. Its advantages include great sensitivity and power. All different subjects of a cluster were equivalently correlated with each other over time. Group × time interactions can illustrate differences in changes over time between two groups. The longitudinal analyses included a mixed-modeling approach. We assessed the fixed effects of group-by-week interactions, intervention group, and study week.

All statistical tests were two-sided with a significance threshold set at 0.05. IBM SPSS Version 21.0 (IBM, Armonk, NY, USA) was used for data analysis.

## 3. Results

### 3.1. Demographic Characteristics and Clinical Measures at Baseline

Of 198 participants assessed for eligibility, 75 declined to participate, and 80 did not fulfill the inclusion criteria (Figure 1). Hence, 43 participants were included and randomized, with 22 allocated to the BLT group and 21 assigned to the DRL group. All participants completed the study and outcome assessments. The baseline demographic characteristics and clinical measures of the subjects are presented in Table 1. In the total sample, the mean age was 45.0 years (SD 14.7), 81.4% of participants were female, and 86.0% of participants had completed senior high school. The mean age at onset of MDD was 36.9 years (SD 14.0), and the mean MEQ-SA score was 47.2 (SD 11.3). In total, 51.2% (*n* = 22) of the patients had recurrent episodes of MDD, 39.5% (*n* = 17) received light therapy during a period with fewer than 12 h of natural daylight, 30.2% (*n* = 13) had the evening chronotype and 58.1% (*n* = 25) had moderate MDD. None of the participants received psychotherapy, which was offered by psychologists during the trial. Before the enrollment, all participants were on stable dosages of their psychotropic medications for at least 4 weeks, and 90.7% (*n* = 39) were on stable dosages of antidepressants for at least 6 weeks (BLT: *n* = 18, DRL: *n* = 21, *p* = 0.11). The mean duration of any previous treatment for the current depressive episode was 10.21 weeks (SD 3.38, range 6─20) among all participants. Randomization was balanced (Table 1).

### 3.2. Pharmacological Treatments during Light Therapy

Light therapy was applied as an add-on treatment to existing pharmacological treatments, and the different regimens used are presented in Table 2. The proportions of prescribed antidepressants were as follows: escitalopram or fluoxetine, 39.5%; venlafaxine or duloxetine, 27.9%; mirtazapine, 11.6%; bupropion, 16.3%; agomelatine, 27.9%. Meanwhile, 18.6, 14.0, 90.7, and 23.3% of patients were receiving aripiprazole, quetiapine, benzodiazepines, and Z-drugs. The prescribed dose ranges were as follows (mg per day): escitalopram (5–20), fluoxetine (20–40), venlafaxine (75–300), duloxetine (60–90), mirtazapine (15–30), bupropion (150–300), agomelatine (25–50), aripiprazole (2.5–5), and quetiapine (100–400). One patient took only 5 mg of escitalopram without adjunctive antidepressive agents. There were no significant differences in the treatment regimen between the groups. Some drugs were not reported in this article such as propranolol, buspirone, Deanxit (flupentixol 0.5 mg/melitracen 10 mg), clotiapine, and very-low-dose trazodone and imipramine.

### 3.3. Adherence to Light Therapy

The adherence to the intervention was supported by the data from the self-report forms. The rate of compliance with light therapy for ≥21 days did not differ between the two groups (95.5% [*n* = 21] in the BLT group vs. 85.7% [*n* = 18] in the DRL group). In the BLT group, the mean start time was 07:42 AM (SD 01:24, range 04:15–10:21 AM). The mean adherence rate of complying with 30 min for 28 days was 92.1% (range 39.3%–100%) in the BLT group. In addition, the Spearman correlation coefficients between the adherence rate and improvement of depression scores (baseline to week 4) in the BLT group were 0.03 (*p* = 0.89) for HAMD-17 and 0.06 (*p* = 0.79) for MADRS. No significant correlations were found between the adherence rate and improvement of depression scores.

### 3.4. Primary and Secondary Outcomes

After 4 weeks of treatment, the participants’ depressive symptoms on all assessment scales used in this study improved in both the BLT and DRL groups. As presented in Table 3, there were no statistically significant differences in depression total scores (HAMD-17, MADRS, PHQ-9, and six subscales of HAMD-17) at weeks 1, 2, and 4 between the two groups (Appendix A). There were also no significant differences in the magnitude of the change of depressive symptoms on all assessment scales between the BLT and DRL groups at weeks 1, 2, and 4. The mean changes of the HAMD-17 score from baseline to week 4 were 6.68 (SD 4.80) in the BLT group and 6.24 (SD 7.25) in the DRL group (*p* = 0.81). The mean changes of the MADRS score from baseline to week 4 were 10.55 (SD 8.18) in the BLT group and 6.67 (SD 10.03) in the DRL group (*p* = 0.17). The mean changes of the PHQ-9 score from baseline to week 4 were 7.41 (SD 5.60) in the BLT group and 5.76 (SD 5.35) in the DRL group (*p* = 0.33).

Regarding remission rates, there were no significant differences between the BLT and DRL groups at the 4-week endpoint (HAMD-17: 13.6% vs. 23.8%; MADRS: 27.3% vs. 28.6%, Table 3). In addition, response rates also did not differ between the groups at this endpoint (HAMD-17: 31.8% vs. 33.3%; MADRS: 40.9% vs. 33.3%; PHQ-9: 36.4% vs. 33.3%).

Using HLM with adjustment for age, gender, and baseline scores, the BLT group displayed no significant decrease of the HAMD-17 score over the study period (weeks 1, 2, and 4) compared to that in the DRL group (Table 4). Similarly, the MADRS and PHQ-9 scores did not significantly differ between the groups during the entire trial. All subscales of HAMD-17 displayed no significant differences between the BLT and DRL groups (Appendix A).

### 3.5. Subgroup Analysis

In the natural daylight <12 h subgroup, the changes of the HAMD-17 and MADRS scores did not differ between the BLT (*n* = 8) and DRL (*n* = 9) groups at weeks 1, 2, and 4. At the 4-week endpoint, the median improvement of scores on the HAMD-17 score in the BLT and DRL groups was 5 (IQR 6) and 5 (IQR 9), respectively, (*p* = 0.63). The median improvement of the MADRS score was 8 (IQR 13) in the BLT group and 6 (IQR 14) in the DRL group (*p* = 0.27).

### 3.6. Side Effects of Light Therapy

At the 4-week endpoint, the most frequently reported side effects were blurred vision, affecting six participants (BLT, *n* = 2; DRL, *n* = 4), and somnolence, affecting five participants (DRL, *n* = 5). There were few significant differences between the groups in the occurrence of adverse events during the treatment sessions. In particular, participants in the BLT group had significantly lower rates of fatigue (0% vs. 19%, *p* = 0.048) and somnolence (0% vs. 23.8%, *p* = 0.021). There was no shifting to hypomania based on a clinical mental-state examination during and at the end of the study.

## 4. Discussion

This RCT indicated that adjunctive BLT was not superior to placebo light in reducing depressive symptoms in a small sample with non-seasonal MDD, as measured using HAMD-17, MADRS, PHQ-9, and HAMD-17 subscale scores and an HLM approach. There were no significant differences between the groups in the response and remission rates. Adjunctive BLT for 4 weeks was not effective for reducing depressive symptoms in a subtropical area with extensive daylight (24.5°–25.5° N). Notably, a high level of participant adherence and completion of outcome assessments were some important features of our study. Since accessing the psychiatrists was convenient in our study area [36], the long-term patient–psychiatrist relationship and assistants’ efforts were also contributing factors.

Consistent with prior findings [37,38], BLT was not efficacious as an adjunctive therapy to antidepressants in patients with non-seasonal MDD in the present study. This observation is also supported by two meta-analyses that did not find robust evidence of efficacy for BLT as an adjunct to the antidepressant medication [6,34]. However, our results contrast with the positive findings of some studies using combinations of light therapy and an antidepressant in comparison to placebo light and an antidepressant. There are possible explanations for these discrepancies. Studies with positive findings had unique features such as the exclusion of patients with treatment-resistance depression [39], the use of a treatment period exceeding 4 weeks [39], approximately 75% of the patients with depression with melancholic features [40], about half of subjects with seasonal worsening without fulfilling the DSM criteria [41], and study periods of autumn and winter [39,40,41]. These studies were also conducted in countries at higher latitudes than our region, such as Canada, Denmark, and Italy. These features possibly enhanced the response to light therapy.

In our study, similar to clinical practice in hospitals, BLT was administered in addition to various psychopharmacological regimens for MDD. Some dosages of the antidepressants administered to our participants were close to the lower limits of the usual dosage, as per the relevant guidelines [9,10,11], but these conditions were consistent with the common treatment dosages for MDD in Asian patients [12]. Several therapeutic strategies were identified in our sample, such as combination regimens and augmentation, which reflected difficulties in optimizing treatment responses for MDD in real practice. Most clinicians would view half of our study participants as having poor drug responses. Generally, patients with depression who do not respond to medications are predicted to exhibit poor symptom outcomes in the future [42]. In previous non-seasonal depression studies, the duration of light therapy ranged from 5 to 90 days (mean 30; median 32) [6]. Therefore, the escalation of treatment duration could be considered for investigating a specific population. Given that SSRIs were used in most trials [6,8], various antidepressive agents other than SSRIs are required for further investigation with light therapy.

The extent of exposure to daylight could affect human well-being. The well-being (total scores for mood, social contact, and energy on the Seasonal Pattern Assessment Questionnaire) was lowest in autumn and winter and highest in spring and summer both for SAD and non-SAD cohorts [43]. In total, 60.5% of our sample received light therapy during a period of at least 12 h of natural daylight per day. The subgroup with fewer than 12 h of natural daylight per day did not display stronger responses to BLT in terms of depression scores. We cannot dismiss the possibility that the effect of individual BLT is limited in people living in regions with extensive daylight and modest seasonal variations. Latitude is associated with daylength variability [31]. Compared to prior studies, our study was conducted at a low latitude with little daylength variability and higher levels of sunshine. Previous studies reported that less daylength variability, greater sunshine experience, and higher ambient temperature are associated with less mental distress and hyperthymic temperament [30,31,44,45,46]. In the present study, the daylength was ≥10 h: 35 m in autumn and winter, and the average ambient temperatures were 22.1 °C from October to December and 19.1 °C from January to March. The climates in autumn and winter are expected to be comfortable for people to perform outdoor activities and be exposed to extensive sunshine. Hence, these climatic factors might explain the low effect of BLT in our study participants. Of note, one study [7], conducted in a climate similar to the climate our trial was conducted in, investigated adjunctive light treatment for depressed evening chronotype subjects (*n* = 93) for 5 weeks, and the between-group effect size was small (Cohen’s *d* = 0.17). Moreover, the small sample size (*n* = 43) and low statistical power in our study, which could lead to type II errors, may explain our negative results. In previous non-seasonal depression studies, the total sample size ranged between 10–102 (mean 47, median 44) [6]; therefore, a larger sample size is needed to investigate the possible low effect of BLT. In contrast, it is possible that a specific population in our study area may benefit from BLT for a short period, such as patients with movement limitations and institutionalized care, who may not be exposed to extensive daylight.

Placebo effects represent another possible explanation for the similar effects of BLT and DRL. There is evidence that the effect of a physical placebo is greater than that of a pharmacological placebo, and that the placebo effect was greater in short-term studies than in long-term ones [47]. The placebo response varied among light therapy studies, ranging from 36–46% [48]. In our study, the response rate to placebo light was close to that of BLT, which was consistent with similar expectation scores in the study groups.

The HAMD score is considered the gold standard for assessing depression severity in clinical trials. However, the factor structure of HAMD is multidimensional and only partly related to the main conception of MDD in DSM-IV [49]. We, therefore, used an additional observer rating scale (MADRS) and self-reported scale (PHQ-9) and further analyzed the subscales of HAMD, i.e., well-known unidimensional subscales (Bech, Maier, Gibbons), and specific subscales (Santen, anxiety, retardation). However, these evaluations did not identify differences in responses between the BLT and DRL groups under HLM.

This study had additional limitations. First, we did not strictly control the drug distributions. This study design probably has good external validity because of its similarity to real-world conditions. Second, we did not instruct the participants to avoid spending excessive time outdoors. The duration of outdoor exposure to natural daylight was not tracked for these outpatients. Third, the enrollment of participants with mild depression (HAMD-17 ≥ 13) who were more likely to spontaneously remit might contribute to nontreatment effects. Nonetheless, consistent with other trials [7,50,51,52], add-on therapy was administered to patients including those with mild depression, implying partial response to antidepressants. Fourth, the majority of our participants had already taken benzodiazepines and Z-drugs for anxiety and sleep symptoms. Fifth, we did not administer a specific questionnaire to detect the changes in subjective sleep experience nor measure sleep patterns under the treatment. Sixth, the timing of morning BLT was not individualized in relation to the MEQ-SA score, which may influence its efficiency [53]. Seventh, we did not examine biological correlates of the circadian rhythm such as salivary melatonin concentrations [54]. Eighth, since BLT might need some time to have an effect on depressive symptoms [55], a follow-up assessment for observing long-term effects would be needed.

## 5. Conclusions

This study’s strength includes being conducted in a subtropical environment (24.5°–25.5° N) on Asians with depression. Since a large majority of studies on BLT have been conducted in the northern hemisphere on patients of white descent, this study examined a diverse, depressed population. The conclusions about the efficiency of adjunctive BLT in our small sample are limited. Nonetheless, our findings show that several factors play important roles in influencing the efficacy of adjunctive BLT, such as patients with poor drug response, different antidepressive regimens, the duration of BLT, and daylength variability (i.e., natural daylight in the environment). Researchers may consider these important factors for non-seasonal depression studies in subtropical environments. Furthermore, an escalation in the BLT duration to investigate a specific population could be considered.

## Figures and Tables

**Figure 1 ijerph-19-12430-f001:**
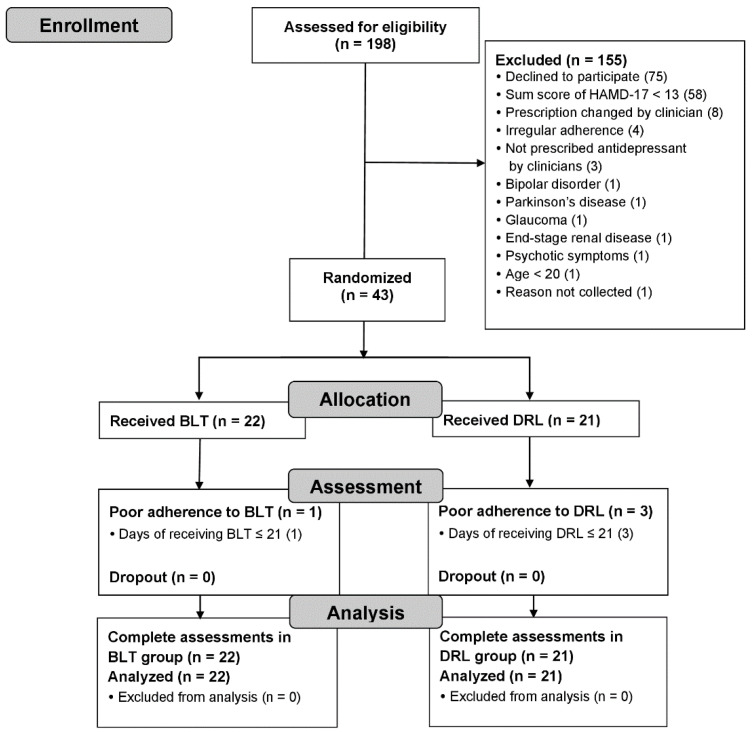
Participant flow diagram through the entire study. BLT: bright light therapy; DRL: dim red light; HAMD-17: Hamilton Depression Rating Scale-17 items.

**Table 1 ijerph-19-12430-t001:** Demographic characteristics and clinical measures at baseline.

	BLT (*n* = 22)		DRL (*n* = 21)		
Characteristics/Measures	Mean	SD	Mean	SD	*p*
Age	47.1	15.0	42.8	14.3	0.34
Age at onset of MDD	39.0	13.9	34.8	14.1	0.34
MEQ-SA	46.9	12.7	47.5	10.0	0.87
Expectation for light therapy	6.41	1.33	5.48	2.11	0.15
Duration of any previous treatment for the current depressive episode (wk)	10.55	3.96	9.86	2.69	0.76
**Characteristics/Measures**	** *n* **	**%**	** *n* **	**%**	** *p* **
**Gender**					1.0
Female	18	81.8	17	81.0	
Male	4	18.2	4	19.0	
**Marriage**					0.67
Married	6	27.3	7	33.3	
Others	16	72.7	14	66.7	
**Highest education level**					0.54
Junior high school/lower	4	18.2	2	9.5	
Senior high school/junior college	10	45.5	8	38.1	
University/higher	8	36.4	11	52.4	
**Employment status**					0.36
Employed full-time	15	68.2	11	52.4	
Others	7	31.8	10	47.6	
**Past major depressive episodes**					0.88
0	11	50.0	10	47.6	
≥1	11	50.0	11	52.4	
**Natural daylight**					0.66
Daylight ≥ 12 h/day	14	63.6	12	57.1	
Daylight < 12 h/day	8	36.4	9	42.9	
**Chronotype**					0.82
Morning/intermediate type	15	68.2	15	71.4	
Evening type	7	31.8	6	28.6	
**Severity of MDD on HAMD-17**					0.29
Mild (13–16)	3	13.6	7	33.3	
Moderate (17–23)	15	68.2	10	47.6	
Severe (≥24)	4	18.2	4	19.0	

BLT: bright light therapy; DRL: dim red-light therapy; HAMD-17: Hamilton Depression Rating Scale-17 items; MDD: major depressive disorder; MEQ-SA: Morningness–Eveningness Questionnaire Self-Assessment; SD: standard deviation.

**Table 2 ijerph-19-12430-t002:** Pharmacologic treatment with antidepressive regimens and BZD/Z-drug.

	BLT (*n* = 22)	DRL (*n* = 21)	
Antidepressive regimens	*n* (%)	*n* (%)	*p*
One AD	11 (50.0%)	10(47.6%)	NS
Combination of two ADs	2 (9.1%)	6 (28.6%)	NS
One AD with ARI	5 (22.7%)	1 (4.8%)	NS
One AD with QUE	4 (18.2%)	2 (9.5%)	NS
Two ADs with ARI	0 (0.0%)	2 (9.5%)	NS
**BZDs and Z-drug**	***n* (%)**	***n* (%)**	** *p* **
BZDs	21 (95.5%)	18 (85.7%)	NS
Z-drug	5 (22.7%)	5 (23.8%)	NS

AD: antidepressant; ARI: aripiprazole; QUE: quetiapine; BZDs: benzodiazepines; Z-drug: zolpidem or zopiclone; NS: not statistically significant.

**Table 3 ijerph-19-12430-t003:** Outcome measures in treating major depressive disorder over time using bright light therapy (*n* = 22) and dim red light (*n* = 21).

Outcome	Wk	Total Score		*p*	Change of Score		*p*	Effect Size ^a^	Response		*p*	Remission		*p*
		BLT	DRL		BLT	DRL			BLT	DRL		BLT	DRL	
		Mean (SD)	Mean (SD)		Mean (SD)	Mean (SD)			*n* (%)	*n* (%)		*n* (%)	*n* (%)	
**HAMD-17**	0	20.59 (4.17)	19.81 (5.76)	0.61										
	1	17.23 (5.32)	14.95 (5.85)	0.19	3.36 (3.11)	4.86 (4.69)	0.82		0 (0)	1 (4.8)	0.49	0 (0)	1 (4.8)	0.49
	2	15.23 (5.48)	14.14 (6.18)	0.55	5.36 (5.01)	5.67 (5.15)	0.85		4 (18.2)	4 (19)	1.00	1 (4.5)	4 (19)	0.19
	4	13.91 (5.90)	13.57 (6.95)	0.86	6.68 (4.80)	6.24 (7.25)	0.81	0.07	7 (31.8)	7 (33.3)	0.92	3 (13.6)	5 (23.8)	0.46
**MADRS**	0	26.09 (6.09)	24.19 (7.08)	0.35										
	1	20.50 (7.50)	18.57 (7.14)	0.39	5.59 (4.94)	5.62 (6.35)	0.99		3 (13.6)	3 (14.3)	1.00	2 (9.1)	3 (14.3)	0.66
	2	18.14 (8.31)	18.05 (8.39)	0.97	7.95 (6.29)	6.14 (6.68)	0.37		4 (18.2)	5 (23.8)	0.72	5 (22.7)	5 (23.8)	1.00
	4	15.55 (9.77)	17.52 (9.45)	0.50	10.55 (8.18)	6.67 (10.03)	0.17	0.42	9 (40.9)	7 (33.3)	0.61	6 (27.3)	6 (28.6)	0.92
**PHQ-9**	0	17.77 (5.01)	16.10 (4.61)	0.26										
	1	13.86 (6.43)	11.76 (3.56)	0.19	3.91 (4.15)	4.33 (3.17)	0.71		3 (13.6)	1 (4.8)	0.61			
	2	11.73 (6.41)	10.81 (4.07)	0.58	6.05 (5.19)	5.29 (4.56)	0.61		8 (36.4)	4 (19.0)	0.21			
	4	10.36 (6.17)	10.33 (4.94)	0.99	7.41 (5.60)	5.76 (5.35)	0.33	0.30	8 (36.4)	7 (33.3)	0.84			

BLT: bright light therapy; DRL: dim red-light therapy; HAMD-17: Hamilton Depression Rating Scale-17 items; MADRS: Montgomery–Asberg Depression Rating Scale; PHQ-9: Patient Health Questionnaire-9; SD: standard deviation; Response was defined as ≥50% reduction in baseline score; Remission was defined as a HAMD-17 score ≤7 or MADRS ≤10; ^a^ Effect size: Cohen’s *d*.

**Table 4 ijerph-19-12430-t004:** Comparison of the effects in treating major depressive disorder for 4 weeks between bright light therapy and dim red-light groups, using hierarchical linear mixed-modeling ^a^ to show group and time effects and interaction of group and time.

Outcome	Group Effect			Time Effect			Group × Time Effect		
	*B*	95% CI	*p*	*B*	95% CI	*p*	*B*	95% CI	*p*
**HAMD-17**	2.67	−0.19 to 5.54	0.07	−0.69	−1.71 to 0.33	0.18	−0.97	−2.39 to 0.45	0.18
**MADRS**	2.64	−1.41 to 6.70	0.20	−0.52	−2.09 to 1.04	0.51	−1.95	−4.15 to 0.24	0.08
**PHQ-9**	1.98	−0.80 to 4.77	0.16	−0.71	−1.57 to 0.15	0.10	−0.42	−2.24 to 0.17	0.09

BLT: bright light therapy; DRL: dim red-light therapy; HAMD-17: Hamilton Depression Rating Scale-17 items; MADRS: Montgomery–Asberg Depression Rating Scale; PHQ-9: Patient Health Questionnaire-9; 95% CI: 95% confidence interval; ^a^ Adjusted for age, gender, and baseline scores.

## Data Availability

The original contributions of this study are in this article. The request for information could be directed to the corresponding author.

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
