# Peer review of "Adjunctive Bright Light Therapy for Non-Seasonal Major Depressive Disorder: A Randomized Controlled Trial"

_ijerph, 2022, doi:10.3390/ijerph191912430_

Round 1

Reviewer 1 Report

In this study adjunctive bright light therapy (BLT) was conducted in adults with non-seasonal depression who live in a subtropical area with extensive daylight and was compared to a dim red light condition. The outpatients received 4 weeks of BLT or red dim light (30 min) in the morning with light boxes adjunctive to antidepressant medication which dosage was stable for 4 weeks before enrollment. Assessments of questionnaires took place at week 1, 2 and 4 of intervention. 

It is a well written study with an interesting research question in this special subtropical area. Strengths of the study are the good external validity (BLT was conducted at home on their own), that the antidepressant medication dosage was stable during the intervention time, that the side effects were documented and that there were no drop-outs to post assessments (good treatment adherence). 

However, there are many limitations. 

First of all, the sample size of 21 or 22 per group was really small compared to the sample size of 41 in each arm which was determined in the power analysis. Thus, conclusions about the efficiency of adjunctive BLT in this sample are limited. 

In addition, timing of BLT in the morning was not individualized in relation to the outpatients chronotype (the MEQ could be used to determine the ideal time slot for BLT). Available evidence shows that individual timing of BLT is a crucial factor for its efficiency (Wirz-Justice et al., 2013). 

Other limitations are the lack of biological correlates such as melatonin and a questionnaire assessing sleep patterns. Since previous studies revealed that BLT might need some time to have an effect on depressive symptoms (e.g. Martiny et al., 2015), a follow up assessment would have been important. 

Due to these limitations, the present study does not add much to the evidence for the effects of BLT in adults with depression.

In addition, the following minor aspects need to be considered: 

Abstract: Please add age of the sample (range and mean).

Introduction: Please explain the presumed mechanisms of BLT. 

Introduction and Methods: Since first-line treatments of depression include psychotherapy, I miss informations about it. Was ongoing psychotherapy an exclusion criteria?

Statistical analysis: Why did you use an effect size of a previous (similar) BLT study for the power analysis?

Sample description: Please add informations about the severity of depression diagnosis (mild/moderate/severe). A meta-analysis revealed that BLT had better efficiency in outpatients with milder depressive symptoms (Tao et al., 2020). 

Figure 1/Flow Chart: Why Follow-Up? It is just a post assessment, or?

References:

Wirz-Justice, A., Benedetti, F., & Terman, M. (2013). Chronotherapeutics for affective  disorders: A clinician’s manual for light and wake therapy (2nd ed.). Basel: Karger.

Martiny, K., Refsgaard, E., Lund, V., Lunde, M., Thoughaard, B., Lindberg, L., & Bech, P. (2015). Maintained superiority of chronotherapeutics vs. exercise in a 20-week randomized follow-up trial in major depression. Acta Psychiatrica Scandinavica131,
446–457.

Tao, Long; Jiang, Rui; Zhang, Kuo; Qian, Zhikan; Chen, Peng; Lv, Yili; Yao, Yuyou (2020): Light therapy in non-seasonal depression: An update meta-analysis. In: Psychiatry research 291, S. 113247. DOI: 10.1016/j.psychres.2020.113247.

Reviewer 2 Report

The authors present a randomized trial of 4 weeks of adjunctive Bright Light Therapy (BLT) vs placebo light therapy in patients being treated for Major Depression of at least 6 weeks duration with a HAMD-17 score greater than 12. There was no evidence for improvement in this score or any secondary measure due to BLT.

This trial appears to have enrolled a diverse range of subjects, ranging from patients just beginning treatment to patients on extended treatment who have not responded (please clarify: the inclusion criteria include episode length of 6 weeks or greater). I am wondering if the duration of previous treatment can be clarified and if any subgroup analysis is warranted. This design makes the study a real-world trial whose results can be generalized, but it remains possible that a specific subgroup of patients may benefit from BLT (I would favor additional discussion of this aspect of the trial).

The trial is unique in being conducted at a latitude of 25 degrees. This is similar to the latitude of Miami, Florida, in the United States: it is considered that Seasonal Affective Disorder is very rare at that distance from the equator. The intervention studied in this trial may not have been effective because all subjects were already receiving relatively long photoperiod light throughout the year.

This is a well-conducted trial, with appropriate analysis. The manuscript is clear and well-written. While I wish the results had shown an effect of BLT, this study is an important contribution. I believe it should be accepted for publication with small additions as noted above.
